# Impact Effect of Methyl Tertiary-Butyl Ether “Twelve Months Vapor Inhalation Study in Rats”

**DOI:** 10.3390/biology9010002

**Published:** 2019-12-20

**Authors:** Osama M. Sarhan, Antrix Jain, Hamed M. A. Mutwally, Gamal H. Osman, Sung Yun Jung, Tawfik Issa, Mohamed Elmogy

**Affiliations:** 1Biology Department, Faculty of Science, Umm Al-Qura University, Makkah 673, Saudi Arabia; sarhanomm5975@gmail.com (O.M.S.); pmmutwally@uqu.edu.sa (H.M.A.M.); 2Zoology Department, Faculty of Science Fayoum University, Fayoum 63514, Egypt; 3Advanced Technology Cores, Baylor College of Medicine, Houston, TX 77030, USA; antrixj@bcm.edu; 4Microbial Genetics Department, Agricultural Genetic Engineering Research Institute (AGERI), Giza, Cairo 12619, Egypt; 5Research Laboratories Center, Faculty of Applied Science, Umm Al-Qura University, Mecca 24381, Saudi Arabia; 6Department of Biochemistry and Molecular Biology, Baylor College of Medicine, Houston, TX 77030, USA; syjung@bcm.edu; 7Department of Ophthalmology, Baylor College of Medicine, Houston, TX 77030, USA; Tawfik.Issa@bcm.edu; 8Entomology Department, Cairo University, Giza 12613, Egypt; osmangamal@hotmail.com

**Keywords:** MTBE, LC-MS/MS analysis, histopathology, cancer biomarker

## Abstract

**Simple Summary:**

The detection of major blood-based markers for cancer requires expensive blood tests. Therefore, finding simple and effective blood-based markers is of great interest. The present results suggested that CA II, CA I, and peroxiredoxin2 could be utilized as potential biomarkers for the early detection of tracheal and lung cancer.

**Abstract:**

We investigated the early risk of developing cancer by inhalation of low doses (60 µL/day) of methyl tertiary butyl ether (MTBE) vapors using protein SDS-PAGE and LC-MS/MS analysis of rat sera. Furthermore, histological alterations were assessed in the trachea and lungs of 60 adult male Wistar rats. SDS-PAGE of blood sera showed three protein bands corresponding to 29, 28, and 21 kDa. Mass spectroscopy was used to identify these three bands. The upper and middle protein bands showed homology to carbonic anhydrase 2 (CA II), whereas the lower protein band showed homology with peroxiredoxin 2. We found that exposure to MTBE resulted in histopathological alterations in the trachea and the lungs. The histological anomalies of trachea and lung showed that the lumen of trachea, bronchi, and air alveoli packed with free and necrotic epithelial cells (epithelialization). The tracheal lamina propria of lung demonstrated aggregation of lymphoid cells, lymphoid hyperplasia, hemorrhage, adenomas, fibroid degeneration, steatosis, foam cells, severe inflammatory cells with monocytic infiltration, edema, hemorrhage. Occluded, congested, and hypertrophied lung arteries in addition, degenerated thyroid follicles, were observed. The hyaline cartilage displayed degeneration, deformation, and abnormal protrusion. In conclusion, our results suggest that inhalation of very low concentrations of the gasoline additive MTBE could induce an increase in protein levels and resulted in histopathological alterations of the trachea and the lungs.

## 1. Introduction

Methyl tertiary-butyl ether (MTBE) is a colorless, hydrophilic, flammable liquid that is used as a fuel additive. MTBE is blended to gasoline at 15% volume to obtain a high octane rating and reduce the tailpipe emissions of carbon monoxide and other hydrocarbons [1]. The British Colombia Ministry of Environment guidelines report that the concentration of MTBE in water considered safe for human drinking, freshwater and marine life, and watering livestock ranges from 0.02 to 11 mg/L. Meanwhile, concentrations of MTBE toxic to the growth of plants and animals of the terrestrial or aquatic environment are determined as 4.8 mg/L and 2.5 to 2.9 mg/L, respectively. The LC_50_ of MTBE ranges from 44 to >10,000 mg/L for the survival of invertebrates and vertebrate animals [2]. Moreover, workers at the gasoline stations are constantly exposed to MTBE vapors through direct skin contact, inhalation, or ingestion. In such individuals, MTBE has been detected in the blood, breath, and urine specimens [3]. Exposure to high levels of MTBE causes headaches, nausea, dizziness, and difficulty in breathing [4].

Yang et al. [3] reported that inhalation of MTBE (100–300 µg/mL) induced lipid disorders and liver dysfunctions in animal models. The oral LC_50_ for MTBE was found to be 3866 and 4000 mg/kg for rats and mice, respectively [5]. Actually, liver adenomas were reported in female mice exposed to high doses of MTBE (3000–8000 ppm) by inhalation [6]. Meanwhile, Moser et al. [7] proposed that 8000 ppm of MTBE could cause hepatocarcinogenic effects on CD–1 and B6C3F1 mice. Similarly, Moser et al. [7] reported that MTBE promoted the formation of liver tumors in female mice. A carcinogenicity study of MTBE revealed that male Wistar rats exposed to high doses of MTBE in drinking water for 2 years developed severe and progressive nephropathy and brain astrocytomas as compared to female rats [1]. Furthermore, Clary [8] reported that rats exposed to MTBE developed severe nephropathy with hyaline droplets in the kidneys. However, renal tubular cell tumors were observed in male mice following MTBE inhalation [9]. They suggested that at exposure to high doses of MTBE, kidney toxicity evolved into nephrocarcinogenesis, suggesting the related role of alpha–2 microglobulin as a biomarker. Luo et al. [10] found some potential cancer biomarkers for MTBE-induced liver cancer; these biomarkers were separated from 800 protein spots on a 2D-PAGE gel from human sera. Moreover, lymphoimmunoblastic lymphoma in the lungs, leukemias, astrocytomas, and Leydig cell adenomas were described in rats exposed to MTBE [6]. The studies related to inhalation of high concentrations of MTBE reported the development of malignancies, such as liver, ovary, testicles, and kidney tumors, in laboratory animals. Furthermore, renal adenomas, nephropathy, tumors in testes, and toxic effects on spermatogenic cells have also been reported [11]. In addition, the effect of MTBE on blood parameters and histological alterations were reported in the bone marrow and immune system, brain, respiratory tract, and skin. Moreover, Prescott-Mathews et al. [9] suggested the role of alpha2u-N in MTBE (3000 ppm) inhalation-induced renal tumorigenesis in male rats. The early detection of cancer is important to increase the survival of patients and to avoid the progression of cancer. Knight et al. [12] reported that lung cancer was the leading cause of cancer-related deaths in the world. Gildea et al. [13] concluded that the difference between the first diagnosis of lung cancer and the final diagnosis impacted the progressive late stages of disease, with a concomitant increase in the associated treatment cost. Therefore, early diagnosis of lung cancer may decrease the resource use and costs among patients. In veterinary medicine, Tothova et al. [14] used serum proteins for the diagnosis. Naz et al. [15] detected diagnostic protein fraction (55–57 kDa) as a biomarker in benign prostatic hyperplasia. With respect to developing cancer biomarkers, Lou et al. [10] recognized an exceptional metabolic biomarker, N-acetylaspartate (NAA) in patients with lung cancer, which is present in inconspicuous levels in normal lung cells. Lou et al. [10] developed a sensitive NAA blood assay and found that elevated levels of NAA were detected in the blood, where it was detected as a metabolite in patients with lung cancer in comparison with the levels in age-matched healthy controls. Other studies indicated a close relationship between MTBE and malignancies in laboratory animals [16]. However, the detection of majority of blood-based markers for cancer requires expensive blood tests. Over the past decade, carbonic anhydrases (CAs) have been shown to be important mediators of tumor cell pH by modulating the bicarbonate and proton concentrations for cell survival and proliferation. This has prompted researchers to develop methods to inhibit specific CA isoforms and use this as an anti-cancer therapeutic strategy. Of the 12 active CA isoforms, CA IX and XII, have been considered as anti-cancer targets. Other CA isoforms also show similar activity and tissue distribution in cancers; however, these have not been considered as therapeutic targets for cancer. Peroxiredoxin-2 is a protein that in humans is encoded by the PRDX2 gene. This gene encodes a member of the peroxiredoxin family of antioxidant enzymes, which reduces hydrogen peroxide and alkyl hydroperoxides. The encoded protein may play an antioxidant protective role in cells and may contribute to the antiviral activity of CD8 (+) T-cells. This protein may have a proliferative effect and play a role in cancer development or progression [17]. Studying the effect of MTBE inhalation on rats might provide evidence for cancer signs in humans. Considering this, the detection of early diagnostic biomarkers for prospective cancer has become crucial for public health assessment, especially for workers who live in contaminated regions, such as gasoline filling stations and habitations. As a consequence, the present work investigated the possibility of cancer-induced in male Wister rats exposed to low concentrations of MTBE. The objectives of the current work included: (1) to study the effect of MTBE inhalation at a low dose (60 µL/day for 3, 6, and 12 months); (2) to find cost-effective cancer biomarkers for early detection; and (3) to confirm the prospective cancer potency in the trachea and lungs.

## 2. Materials and Methods

Methyl tertiary-butyl ether was purchased from Sigma Aldrich, Cairo, Egypt (CAS no. 1634–04–4).

### 2.1. Animals

All experiments were conducted using 60 adult male Wistar albino rats weighing 150 to 180 g. These were divided into four groups with 15 rats in each group. Rats were purchased from King Abdulaziz University, Jeddah, KSA. They allowed to adapt for one week under controlled conditions (at 22–25 °C and 50%–55% humidity) before starting experiments at the animal house of the Faculty of Applied Science, Biology Department, Umm Al-Qura University, Makkah, KSA. The Institutional Animal Care and Use Committee approved the protocols for the present study. The animals were used according to the institutional ethical guidelines (Table 1).

### 2.2. Blood and Tissue Samples

After the exposure period, five rats were selected randomly, sacrificed (under light ether anesthesia), and dissected. The blood samples were obtained in heparinized tubes by cardiac puncture. After centrifugation at 4000 rpm for 15 min at 4 °C, blood sera were collected and kept frozen at −80 °C until electrophoresis analysis. Simultaneously, the tracheae and lungs were extracted and processed routinely for paraffin sectioning according to the method described by Bancroft and Gamble [18,19]. After extraction, these organs were immediately washed in cold physiological saline, fixed in 10% formalin buffer for 48 h, dehydrated in an increasing concentration of ethanol, cleared in xylene, embedded in paraffin blocks, followed by cutting into 3 to 5 micron thick sections. The obtained paraffin sections were then de-paraffinized, hydrated, and stained with hematoxylin and eosin (H&E) to monitor the histopathological changes.

### 2.3. Sodium Dodecyl Sulfate-Polyacrylamide Gel Electrophoresis

Rat blood sera (10 µg) from each sample were separated on 10% sodium dodecyl sulfate-polyacrylamide gel electrophoresis (SDS–PAGE). The total protein of the isolates was determined and protein analysis was performed using 10% SDS-PAGE. Protein samples were reduced by boiling for 5 min in loading buffer containing 5% β-mercaptoethanol. Next, the samples were centrifuged at 10,000× *g* for 3 min and directly loaded onto the gel. Protein electrophoresis was performed in vertical sub-cells (Bio-Rad, Watford, UK). Slab gels, containing 10% (*w/v*) resolving gel and 5% stacking gel concentrations of acrylamide, were run at a constant voltage of 80 V for 2 h. Proteins on PAGE gels were fixed in 45% methanol and 10% acetic acid in distilled water and stained in 0.25% Coomassie brilliant blue R-250 (previously dissolved in 10% acetic acid, 50% methanol, and water). The gels were destained for 3 h in 5% methanol and 7% acetic acid in distilled water to visualize protein bands. These bands were photographed in the lightbox according to the standard technique described by Laemmli [20].

### 2.4. LC-MS/MS Analysis

Individual gel pieces were destained and subjected to in-gel digestion using trypsin (T9600; GenDepot, Houston, TX, USA). The tryptic peptides were re-suspended in 10 μL of 5% methanol containing 0.1% formic acid solution and subjected to nanoflow LC-MS/MS analysis on a nano-LC 1000 system (Thermo Scientific) coupled to Q-Exactive Plus Hybrid mass spectrometer from Thermo Scientific. The peptides were initially loaded on a pre-column (2 cm × 100 µm), followed by a 5 cm × 150 µm-analytical columns packed with ReproSil-Pur Basic C18 equilibrated in 0.1% formic acid/water. The peptides were eluted using a 35-min discontinuous gradient of 4% to 26% acetonitrile/0.1% formic acid at a flow rate of 800 nL/min. The eluted peptides were directly electro-sprayed into the mass spectrometer operated in the data-dependent acquisition mode, thereby acquiring higher-energy collisional dissociation (HCD) fragmentation spectra of the top 20 strongest ions. The raw files with MS/MS spectra from the mass spectrometer were searched against target-decoy *Rattus norvegicus* reference sequence database (released January 2016, containing 60,109 entries) in Proteome Discoverer 1.4 interface (Thermo Fisher) via Mascot algorithm (Mascot 2.4, Matrix Science). The following parameters were used for the search: variable modification of oxidation on methionine and protein N-terminal acetylation; 20 ppm precursor mass tolerance; 0.5 Da fragment mass tolerance; and two missed cleavages. The peptides were identified at a 5% false discovery rate (FDR). The calculated area under the curve of peptides was used to calculate iBAQ for protein abundance using the method described by Sung et al. [21].

## 3. Results

Microscopic investigations revealed histopathological alterations in the trachea (Table 2) and the lungs (Table 3) exposed to low doses of MTBE (60 µL) throughout the experimental periods (3, 6, and 12 min of MTBE exposure) in comparison with the trachea and the lungs in the control rats. Table 2 and Table 3 summarize the histological anomalies.

The control trachea (Figure 1a) showed an intact inner normal mucosal layer (Mu) formed of ciliated pseudostratified columnar respiratory epithelium (RE) and lamina propria (LP). Furthermore, the control rats showed normal hyaline cartilage (HC), normal architecture of peritracheal CT containing typical blood vessels (BV) “veins and arteries” with tunica intima (TI), tunica media (TM), and tunica adventitia (TA).

After 3, 6, and 12 min of exposure to MTBE, the trachea showed noticeable alterations. The lining RE showed thickening (Th), deciliation (Dc), desquamation (Ds), mucosal ulceration (MUl), hyperplasia (Hp), and polyp formation (Figure 1b–d). Moreover, flattening (FE) was observed in the lining columnar epithelium similar to that present in the squamous epithelium “metaplasia” (Mp) (Figure 1b–d and Figure 2d), as well as degenerative epithelium (DE) (Figure 2c,d). After 12 min of exposure to MTBE, the tracheal lumen (Lu) revealed the presence of eosinophilic debris formed by necrotic epithelial cells (Nc) and epithelialization (Epi) (Figure 2a,b,d–f) (Table 2).

The CT of LP, after 3 min MTBE exposure, displayed congested blood vessels (CB) and blood capillaries (cc), edema (e), and infiltration of inflammatory cells and lymphoid cells (LC) (Figure 1b–f and Figure 2a–f) with the formation of lymphoid hyperplasia (LH), focal (H), and diffuse hemorrhage (DH) (Figure 2a–f). Certain degenerative changes, such as fibroid changes (Fi), formation of lipid-laden macrophages “foam cells (FC),” small rounded and large fluid-filled acini (Ac) “tracheal adenomas (TA),” appeared along with deeply aggregated acini (Figure 2a,d–f). These structures were formed of small adenomatous corpuscles and were lined with simple glandular cuboidal epithelium (Figure 1e).

The hyaline cartilage (HC) after 3 min of exposure indicated mitotic figures (MF) (Figure 1b), deformation (Df) (Figure 1c), degeneration (DHC) with the presence of some chondroblasts (Ch) (Figure 2b), moderate enlargements (EC), increased diameter (Di) “hypertrophy,” and broadening with protrusions (PC) toward the outer and inner directions with perichondrial thickening (PeT) (Figure 1e,f and Figure 2e,f) (Table 3). Moreover, Figure 1e,f showed perichondrium thickening.

The peritracheal CT revealed congested blood vessels (CB), inflammatory cell infiltration (If), edema (e), monocytic infiltration (M), fibroid changes (Fi), and formation of foam cells (FC), and fatty degeneration (FD) (Figure 1d and Figure 2a,c–e). Furthermore, the thyroid gland showed degenerated thyroid follicles (DT) (Figure 2a) to a certain extent.

The lungs in the control rats (Figure 3a) displayed undamaged bronchioles, blood vessels (BV), and air alveoli (AA). However, the lung tissue exposed to MTBE presented deterioration that increased with the exposure time throughout the experiment. After 3 and 6 min, the lung tissues revealed dilated blood vessels (DV), congested blood vessels (CB) and blood capillaries (cc), and dilated bronchioles (DB) (Figure 3b–f). The lining epithelium of lung bronchioles displayed polyp formation (PF) and metaplasia (Mp) (Figure 3d–f). After 12 min of exposure, more alterations, such as thickening (Th), metaplasia (Mp), and hydropic degeneration (HD), were observed (Figure 4a–c), along with the deciliation (Dc), metaplasia (Mp), and epithelialization (Epi) in the lining epithelia of both bronchioles and air alveoli.

Furthermore, numerous degenerative changes, such as emphysematous changes (Em) with septal destruction of air alveoli (SD) and the incidence of degenerated alveoli (Da) lung adenoma (LA), were noticed (Figure 4c,d). Additionally, collapsed alveoli (CA) were observed along with the formation of large focal abscesses with central liquefaction (CL) surrounded by pyogenic membrane (PM) (Figure 4e,f).

The LP of the lung bronchioles (Br) (Figure 4a), peribronchioles (Pb), and perialveolar (Pa) CT showed infiltration of inflammatory cells (Figure 3c–e and Figure 4a–e) and formation of lymphoid hyperplasia (Figure 3d and Figure 4a–e).

Moreover, the lung parenchyma showed edema (e) in the perivascular (Pv) and peribronchiolar (Pb) regions (Figure 3d and Figure 4e), congested blood vessels (CB), congested blood capillaries (cc) (Figure 3b–f and Figure 4e), hemorrhage (H) (Figure 3c), diffuse hemorrhage (DH) (Figure 4d–f), and fibroid changes (Fi) (Figure 3d,f, and Figure 4e). Furthermore, the formation of foam cells (FC) (Figure 4a,c), steatosis (St) (Figure 4c,e), and the presence of occluded and hypertrophied lung arteries (HA) were observed after 12 min of exposure to MTBE (Figure 4a,c,f). Figure 4a,c illustrates hypertrophied arterial (HA) muscles “tunica media.” Similarly, other blood vessels showed desquamation (DS) in tunica intima and thickening of the outer sheath “tunica adventitia” (Figure 4e). However, the lower panel of Figure 4d,e reveals fibroid changes (Fi) with fibrocytes (F), edema (e), severe emphysema (SE) with epithelialization (Epi), as well as degenerated air alveoli (Da) invaded with numerous monocytes (M).

### 3.1. SDS-PAGE

The total protein profile from blood sera revealed significant qualitative and quantitative differences (Table 4). Scanning analysis revealed major differences in the banding pattern of the proteins. The electrophoretic separation of blood sera collected from male rats after 3, 6, and 12 months revealed three protein bands of 29, 28, and 21 kDa, with relative mobility (RF) of 57, 0.61, and 0.72, respectively. The first, second and third fragments appeared after three months and continued to exist in the samples collected at 6 and 12 months. Thus, we conclude that proteins corresponding to these bands were up-regulated following MTBE inhalation, as revealed by the intensity of the protein bands stained with Coomassie brilliant blue (Figure 5A,B and Table 4, Table 5 and Table 6).

The obtained results suggest that very low concentrations of MTBE inhalation induced the expression of three proteins in the blood sera of rats. The SDS-PAGE analysis of blood sera proteins provided useful information. However, further experiments are required to obtain a clear picture and therefore, we decided to use mass spectroscopy.

### 3.2. LC-MS/MS Analysis

The most abundant proteins from upper, middle, and lower bands (Figure 5) were identified as carbonic anhydrase 2 and carbonic anhydrase 1; carbonic anhydrase 2; and peroxiredoxin 2, respectively, by mass spectrometry (Table 6).

## 4. Discussion

The present study performed a preliminary analysis of the differences in the protein profiles obtained from the blood sera collected at different time points from male rats exposed to low concentrations of MTBE using the polyacrylamide gel electrophoresis technique. SDS-PAGE has been previously and successfully used to detect a novel, blood-based prostate cancer marker [22,23]. Carbonic anhydrases (CAs) have been studied for over 90 years [24]. Since their discovery in 1933, CAs have been at the forefront of scientific studies, ranging from basic enzymology to the application of structural biology and in silico approaches to study protein dynamics [25,26]. Of the 15 isoforms of CAs expressed in humans, only CA IX and CA XII have been implicated in cancer. These enzymes are transmembrane proteins in which their extracellular domains contain the catalytic activity; these are involved in regulating the tumor microenvironment [3,27]. The CA family has been shown to create and maintain the pH differential in tumor cells [28,29,30,31,32,33]. This suggests that CA I could potentially serve as a plasma biomarker [34]. Recently, CA1 gene amplification was detected in approximately 25% of breast cancer studies [32]. Similar to CA1, the most common gene alterations in CA 2 were amplifications, especially in neuroendocrine prostate cancer, breast cancer, prostate adenocarcinomas, and metastatic cancers followed by mutations [32,33]. Thus, both CA I and II may be potential targets for treating several cell-type-specific cancers. The microscopic investigation verified the deteriorating histopathological alterations in rat trachea and lungs of rats at 3, 6, and 12 months after MTBE exposure. Conspicuous histopathological alterations were reported. The lining epithelia of trachea and lung bronchioles showed ulceration, desquamation, hyperplasia, and polyp formation. However, lymphoid hyperplasia, tracheal and lung adenoma, edema, and monocytic infiltration in the tracheal LP and in the perivascular and peribronchiolar CT of bronchioles of lungs, as well as a focal abscess in the lung parenchyma were observed. The reason for the observed hydrops in the epithelial cells could be related to the absorption of water. The fibroid degeneration could be explained by the degeneration of lung parenchyma, activation of fibroblasts with subsequent fibrosis, and defectiveness in the smooth muscle fibers of interstitial CT as a benign smooth muscle tumor ”leiomyoma”. The observed lung leiomyoma might be a consequence of collapsed air alveoli and occlusion of some bronchioles. Similarly, Nuovo and Schmittgen [34] suggested that leiomyomas signified independent and multifocal smooth muscle proliferation. Besides, the observed hyaline cartilage deformation, thyroid follicles degeneration, and lung abscess were not reported earlier following MTBE exposure. With respect to the mentioned adenoma in the trachea and lung tissues, the present investigation suggested that it may be an early stage of adenocarcinoma. In rats and mice, Bird et al. [35] observed hepatocellular adenomas in females, and interstitial cell adenomas of the testes were observed in males exposed to 3000 and 8000 ppm of MTBE. Moreover, testicular and renal adenomas, in addition to tubular carcinoma, were reported in male rats exposed to MTBE. Narrowing of the vascular lumen induces vascular occlusion in the adjacent regions, whereas the hypertrophy of lung arteries may be induced by hyperplastic and hypertrophied smooth muscle fibers of tunica media and increased number of interstitial elastic fibers. The present investigation suggested that the hypertrophy of tunica media could be a result of pulmonary hypertension, inflammation, and partial obstruction or interruption of the blood flow. Structurally, the enlargement of muscle cells could be attributed to the decrease in the nucleo-cytoplasmic ratio induced by increasing the density of sarcoplasmic reticulum and mitochondria. Moreover, Gabella [36] added that the observed hypertrophy could be due to the fact that muscle cells may be dividing longitudinally. With respect to the monocytic infiltration observed in the present study, Li et al. [37] proved that radiolabeled monocytes migrated into the acute inflamed rat lungs. However, Azoulay et al. [38] concluded leukemic pulmonary infiltration to be the first manifestation of acute monocytic leukemia. The hydropic degeneration and lipid-laden macrophages “foam cells” detected in the present work could be explained by the work of Romero et al. [39], who reported that foam cells developed from neighboring alveolar epithelial type II cells, which responded to the injury by accumulating lipids into the distal airspaces of the lungs. Moreover, MTBE inhalation induced histopathological hazards in the brain, liver, spleen, lungs, kidney epididymis, and gonads, as well as led to renal and gonadal tumors in males [40,41]. Furthermore, other reports concluded that MTBE administered in drinking water or by inhalation induced chronic progressive nephropathy of minimal to mild severity in male Wistar rats and Fischer rats “F344”. However, Kenneth et al. [11] reported that MTBE gavage in rat and mouse inhalation elevated tumor rates in the kidneys and Leydig interstitial cells of male rats and leukemia/lymphomas in female rats. They believed that MTBE or its metabolites are cytotoxic, tumorigenic, and could damage DNA, which may increase the cancer potency, unless at a very low dose of 0.003 mg/kg b.wt. Moreover, Belpoggi et al. [42] reported that MTBE exposure induced an increase in dysplasias and immunoblastic lymphomas among male and female rats. In addition, it resulted in reactive hyperplasia of the peribronchial lymphoplasma cellular tissue, also found to be the most frequent alteration in the lungs. Besides, lymphoid hyperplasias were actually explained as reactive marginal zone B-cell lymphomas [42]. Moreover, Schoeb et al. [42] reported that rats exposed to MTBE developed lympho-immunoblastic lymphoma. Recently, Schoeb et al. [42] confirmed that MTBE induced several biochemical and physiological dysfunctions. She suggested that MTBE generated reactive oxygen species with subsequent harmful effects on different body systems and organs as well as on the normal body physiology, including liver, kidneys, blood, bone marrow, and the nervous, respiratory, reproductive, and immune systems. The present investigation demonstrated that inhalation of MTBE at low doses induced a noteworthy difference in the pattern of protein bands in the rat sera. The effect of MTBE has been investigated in blood samples, breath, and urine specimens [18]. Unfortunately, no data have been reported on the effects of MTBE exposure on electrophoretic protein profile of rat blood sera. The present electrophoretic protein profile of blood sera in male Wister rats exposed to MTBE showed the presence of three protein bands. Some of these proteins are identified using mass spectroscopy in direct or indirect association with cancer potency. Numerous investigations revealed that a high dose of MTBE significantly exerted toxicity on white blood cell count, including lymphocytes, granulocytes, eosinophils, as well as on hemoglobin. Moreover, the biochemical parameters, such as triglycerides, cholesterol, and transaminases (ALT and AST), were increased in the sera of treated rabbits [12]. In the literature, there are no reports on the abnormal electrophoretic protein profile of rat blood sera after MTBE exposure to detect early bio-indicators for carcinogenicity in mammalian models. The electrophoretic profile obtained in the present study revealed three abnormal protein bands containing numerous types of proteins. Some of these proteins were identified to be associated with several carcinomas. In conclusion, the present study identified specific protein bands that could be used to develop early, serum-based, cost-effective, diagnostic biomarkers for carcinogenicity. We believe that these biomarkers would assist in early health assessment of workers in environments such as gas stations using gasoline mixed with MTBE and the residents exposed to car fumes. We considered all isoforms of CAs and their possible roles in tumorigenesis as well as their potential as targets for cancer therapy.

## 5. Conclusions

Collectively, our results suggest that SDS-PAGE and mass spectroscopy could serve as potential techniques for analyzing blood-based early diagnostic markers following exposure to very low concentrations of MTBE. These could serve as promising tools for assessing inhalation toxicity and carcinogenicity associated with MTBE vapor inhalation effects. Whether our detected and identified early markers are valid in human blood plasma or not will be the matter of our next investigation. Based on the findings of the present study, we conclude that inhalation of very low concentrations of the gasoline additive MTBE could induce an increase in protein levels much before the first appearance of histopathological effects on male rats.

## Figures and Tables

**Figure 1 biology-09-00002-f001:**
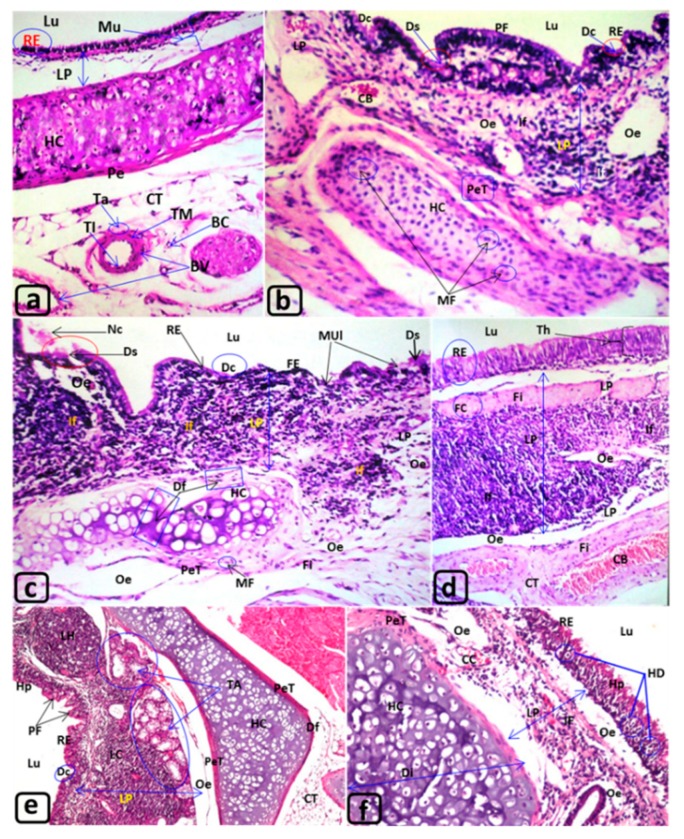
T. S., Trachea of the rat: (**a**) G1 “control”; (**b**–**d**) G2 exposed for 3 months; (**e**,**f**) G3 exposed for 6 months. H&E. (Table 2). Lu: Lumen of control trachea clear from abnormal secretion of any cellular debris. RE: The lining tissue called mucosa (Mu), formed of respiratory epithelium (RE) and the underneath connective tissue, lamina propria (LP). HC, CT and BV: The submucosal layer involved hyaline cartilage (HC) that covered with perichondrium (Pe) sheath and the peritracheal CT surrounded the trachea with numerous blood vessels (BV) that showed normal tunica intima (TI), tunica media (muscularis) (TM), and tunica adventitia (Ta) consisted of substantial sheath of CT, as well as normal blood capillaries (BC). **BV:** blood vessel; **CT:** peritracheal connective tissue; **HC:** hyaline cartilage; **LP:** lamina propria; **Lu:** lumen of trachea; **RE:** respiratory epithelium. Magnifications ×40. H&E. **Figure 1b–d: trachea of rats (G2) exposed to MTBE at 60 µL/3 min/day, for 3 months. Figure 1b–d: Lu:** Lumen (Lu) showed the presence of eosinophilic debris formed from the necrotic epithelial cells (Nc), Figure 1c. RE: The respiratory epithelium (RE) displayed deciliation (Dc), desquamation (Ds), polyp’s formation (PF) mucosal ulceration (MUl), which deformed into flattened squamous epithelium (FE) “metaplasia” (Figure 1b,c) and/or thickening (Th) “hyperplasia” (Figure 1d). LP: demonstrated congested blood vessel (CB); diffuse accumulation of inflammatory cells infiltrations (If), edema (Oe) (Figues 1b–d), as well as, fibroid changes (Fi) with the presence of few foam cells (FC) (Figure 1d). HC: indicated mitotic figures (MF) with thickened perichondrium (PeT) (Figure 1b) and deformation (Df) (Figure 1c). CT and BV: the peritracheal CT illustrated fibroid changes (Fi) besides severs dilatation and congestion of some blood vessels (CB). Magnifications (Figure 1b–d: ×40). H&E. **Figure 1e,f: trachea of rats (G3) exposed to MTBE at 60 µL/3 min/day, for 6 months.** Lu: showed, also, the presence of eosinophilic debris and necrotic epithelial cells, but not shown in the present Figure 1e,f. RE: showed deciliation (Dc), hyperplasia (Hp), polyp’s formation (PF) (Figure 1e), and hydropic degeneration (HD) (Figure 1f). LP: more alteration was observed such as: aggregation of lymphoid cells (LC), lymphoid hyperplasia, and accretion of tracheal adenomas (TA) (Figure 1e) and edema (Oe) (Figure 1e,f) with dilatation and congestion in the blood capillaries (CC) (Figure 1f). HC: LP: more alteration was observed such as: aggregation of lymphoid cells (LC), lymphoid hyperplasia, and accretion of tracheal adenomas (TA) (Figure 1e) and edema (Oe) (Figure 1e,f) with dilatation and congestion in the blood capillaries (CC) (Figure 1f). presented deformation (Df) due to outward protrusion with perichondrial thickening (PeT) (Figure 1e,f) and increased its diameter (Figure 1f). CT and BV: The peritracheal CT showed the same alteration observed in Figure 1b–d, not shown in the present Figure 1e,f. Magnifications (Figure 1e, ×16 and 1f ×40), H&E.

**Figure 2 biology-09-00002-f002:**
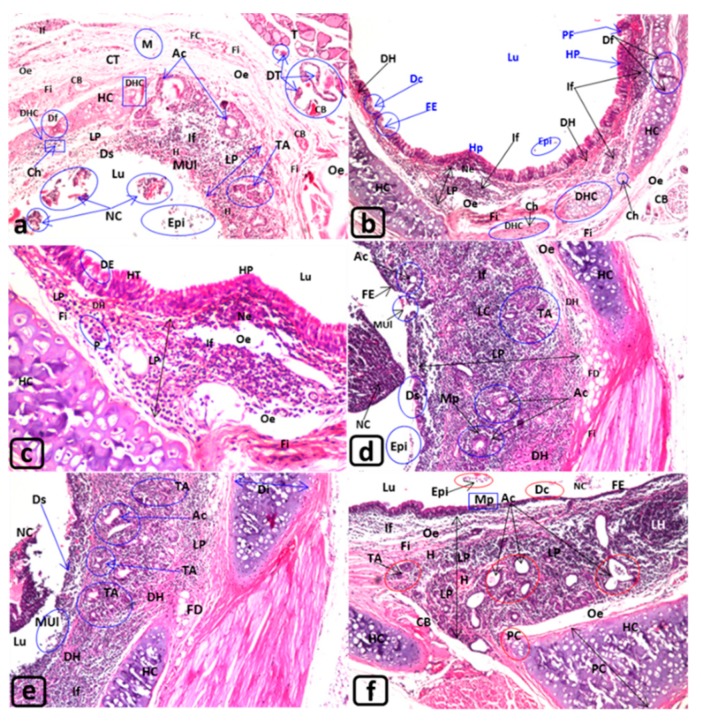
**T. S., Trachea of the rat: (a–f) G4 exposed for 12 months. H&E. (Table 2). Figure 2a–f: trachea of rats (G4) exposed to MTBE at 60 µL/3 min/day, for 12 months.** Lu: illustrated epithelialization (Epi) and necrotic cells (NC) (Figure 2a,b,d–f). RE: showed severe desquamations (Ds) and mucosal ulceration (MUl) (Figure 2a,d,e), degeneration (DE) and hypertrophy (HT) (Figure 2e), and hyperplasia (HP) (Figure 2b,c) or flattened epithelia (FE) “metaplasia” (Figure 2b,d,f). LP: showed increase in size (Figure 2a–f: see bi-head arrow), focal hemorrhage (H) (Figure 2a,f), diffuse hemorrhage (DH) (Figure 2b–e), edema (Oe) (Figure 2a–d,f), fibroid changes (Fi) and inflammatory cells infiltrations (If) (Figure 2a–f), lymphocytic cell infiltration (LC) (Figure 2d), lymphoid hyperplasia (LH) (Figure 2f), furthermore, a small necrotic area (Ne) (Figure 2b,c), fatty degeneration (FD) (Figure 2d,e) with fibroid changes (Fi) (Figure 2a–d,f) and aggregation of plasma cells (P) (Figure 2c). Also, the LP revealed the formation of tracheal adenomas (TA) (Figure 2a,b,d,e) and some of them fused or grown to form large filled fluid large acini (Ac) (Figure 2a,d–f), with metaplasia (Mp) in their walls (Figure 2d). HC: revealed an increase in diameter (Di) or enlarged and protruded (PC) toward the outer and inner directions, but, the cartilage of other trachea displayed deformation (DF) and degeneration (DHC) with the presence of numerous chondroclasts (Ch), which may induce degeneration. CT, BVand T: The peritracheal CT, showed edema (Oe), fibroid degeneration (Fi) (Figure 2a,b) with fibrocytes (Figure 2a), dilated and congested blood vessels (CB) (Figure 2a,b), in addition, inflammatory cells infiltrations (If) and monocytic infiltration (M) (Figure 2a). T: The thyroid the thyroid gland (T) showed few degenerated thyroid follicles (DT) with congested blood vessels (CB) (Figure 2a). Magnifications (Figure 2a ×40; 2b ×16; 2c ×80; 2d–f ×40).

**Figure 3 biology-09-00002-f003:**
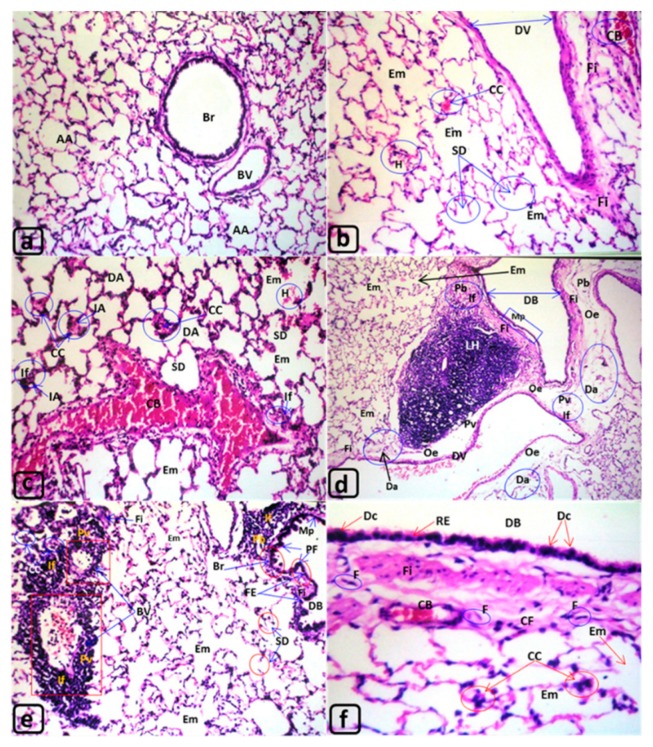
T. S., Lungs of the rat: (**a**) G1 “control”; (**b**,**c**) G2 exposed for 3 months; (**d**–**f**) G3 exposed for 6 months. H&E. (Table 3). **Histological alteration in rat lung (Figure 3 and Figure 4). Figure 3a**: G1, Lung of control rat. Lu: Normal (clear) lumen of lung bronchi “Br” and normal lumen of air alveoli “AA”. RE: Normal respiratory epithelium RE of lung bronchi. AA: Normal simple squamous epithelia of air alveoli “AA”. CT: Normal connective tissue “CT” of interalveolar septa, peribronchiolar “Pb” and perivascular regions “Pv”. BV, CC: Normal blood vessels “BV” and blood capillaries “CC” of the interalveolar septa, Pb and Pv connective tissues “CT”. Magnifications (Figure 3a), H&E. **Figure 3b,c: Lung of rats (G2) exposed to MTBE at 60 µL/3 min/day, for 3 months.** Lu: The lumen of some air alveoli was dilated (DA) with focal hemorrhage “H” (Figure 3c). RE: The respiratory epithelia of air alveoli appeared normal with mild thickening (Figure 3c). AA: The air alveoli “AA” showed dilatation “DA” (Figure 3c) with numerous emphysematous changes “Em” (Figure 3b,c). CT: The connective tissue “CT” of the interalveolar regions showed destructed septa “SD” (Figure 3b,c). Pb and Pv: The connective tissue “CT” of peribronchiolar “Pb” and perivascular “Pv” showed inflammatory cells infiltrations (If) and perivascular fibroid (Fi) changes (Figure 3b,c). BV and CC: The blood vessels illustrated severe dilatation (DV) (Figure 3b), congested blood vessels (CB) (Figure 3b,c), and blood capillaries (CC) (Figure 3c) of some inter-septal “IA” CT of air alveoli. Magnifications (Figure 3b: ×40; 3c: ×80), H&E. **Figure 3d–f: Lung of rats (G3) exposed to MTBE at 60 µL/3 min/day, for 6 months.** Lu: The lumen of some bronchi displayed severe dilatation “DB” (Figure 3d,e). RE: The respiratory epithelium “RE” of dilated bronchi “DB” appeared de-ciliated “Dc” (Figure 3f), shortening or flattened metaplasia “Mp” (Figure 3d) with polyp formation “PF” (Figure 3e). AA: The air alveoli “AA” demonstrated numerous emphysematous changes “Em” (Figure 3d–f) and degenerated alveoli “Da” (Figure 3d), and some lining epithelia showed polyp’s formation (PF) or flattened epithelium “metaplasia” (Mp) (Figure 3e). CT The CT: of inter alveolar septa and peribronchiolar “Pb” region showed inflammatory cells infiltration “If” (Figure 3d,e), fibroid changes “Fi” (Figure 3d–f), with numerous fibrocytes “F” (Figure 3f) and edema “Oe” (Figure 3d). Pb and Pv: Large lymphoid hyperplasia (LH) (Figure 3d) was observed in the Pb and perivascular Pv regions. BV and CC: The blood vessels “BV” (Figure 3e) in the CT of interalveolar, peribronchiolar regions showed dilated “DV” (Figure 3d), congested blood vessel (CB) and blood capillaries (CC) (Figure 3e,f). Magnifications (Figure 3d: ×16; 3e: ×20 and 3f ×80), H&E.

**Figure 4 biology-09-00002-f004:**
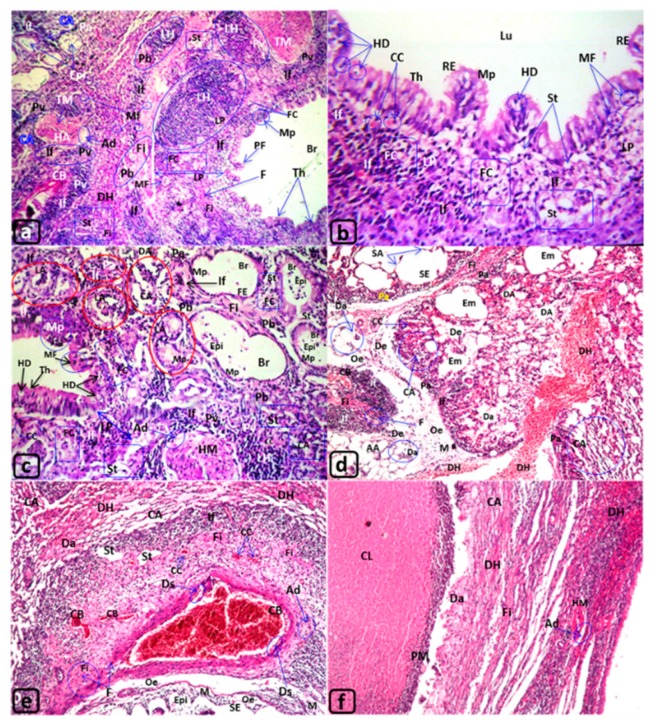
**T. S., Lungs of the rat: (a–f) G4 exposed for 12 months. H&E. (Table 3). Figure 4a–f: Lung of rats (G4) exposed to MTBE at 60 µL/3 min/day, for 12 months.** Lu, RE and AA: The lumen of air alveoli in many regions frequently showed collapsed ““CA” (Figure 4a,c–f), dilated “DA” (Figure 4c,d), severe emphysematous changes “SE” due to destruction or loss their septa “SA” (Figure 4d,e), or degenerated alveoli “Da” (Figure 4d,f). The lining respiratory epithelium “RE” of bronchi seemed flattened metaplasia “Mp” (Figure 4c), thickening “Th”, or grown into polyp’s shape “PF” (Figure 4a,b). Also, the lining epithelia of bronchioles and air alveoli “AA” showed epithelialization “Epi” (Figure 4a,c) or disappeared due to degeneration “Da” of numerous alveoli (Figure 4d,f). Moreover, the lung parenchyma presented an abusive case of severe focal abscess with central liquefaction “CL” that was surrounded by pyogenic membrane “PM” (Figure 4f). CT, Pb and Pv: The connective tissues “CT” of LP, the peribronchiolar “Pb” and perivascular “Pv”, and inter-septal perialveolar “Pa” regions packed with dense inflammatory cells infiltrations ‘If” (e.g., Figure 4a–d), lymphoproliferative hyperplasia “LH” (Figure 4a), monocytic infiltration “M” (Figure 4d,e), numerous lipid-laden macrophages “foam cells” “FC” (Figure 4a,b), lung adenoma “LA” (Figure 4c), mitotic figure “MF” (Figure 4d,e), and steatosis “St” (Figure a,b,e), in addition, edema “Oe” (Figure 4d,e), diffuse hemorrhage “DH” (Figure 4d,e,f), fibroid degeneration “Fi” (Figure a,d–f) with fibrocytes “F” (Figure 4a,e). BV and CC: Congested blood vessels “CB” and capillaries “CC” showed in the connective tissues (Figure 4d,e). On the other hand, some of lung arteries appeared hypertrophied “HA” and occluded arterial lumen with hypertrophy “HM” of tunica media “TM” and tunica adventitia “Ad” (Figure 4a,c,e). However, the lining epithelium of the larger blood vessel appeared occluded, the lumen showed desquamation “Ds” (Figure 4a,e), in addition, hypertrophy “HM” of tunica media “TM” and tunica adventitia (Ad). Magnifications (Figure 4b: ×80; 4d: ×16; 4c,e,f: ×40), H&E.

**Figure 5 biology-09-00002-f005:**
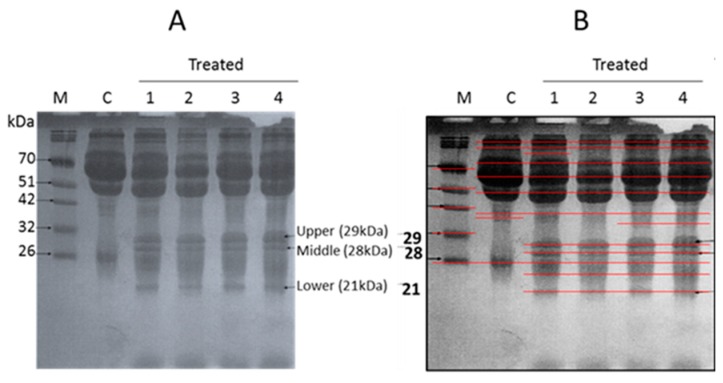
(**A**) Coomassie blue-stained SDS PAGE of sera. Lane M: protein marker, Lane C: control sample. Lane 1 treated sample after 3 months. Lane 2: treated sample after six months. Lane 3 treated sample after 12 months—Lane 4: treated samples after 12 months. The high, middle and lower band were dissected and subjected to MS analysis. (**B**) The image of SDS-PAGE gel was obtained by Bio-Rad gel doc system to detect the bands and deference’s between each lane.

**Table 1 biology-09-00002-t001:** Showing the experimental design for the control and exposed animals.

Experimental Design (Four Groups, n = 15)			
Rat Groups:	MTBE Vapor Dose	Inhalation Time	Experimental Period
G1: control	Non exposed	Non exposed	0–12 months
G2: expose to	60 µL/day	3 min	0–3 months
G3: expose to			0–6 months
G4: expose to			0–12 months

**Table 2 biology-09-00002-t002:** Of Histological Alterations of Rat Trachea Represented in Figure 1 and Figure 2.

Trachea of Rats Exposed to MTBE Vapor Inhalation 60 µL/3 min/day for 3, 6, or 12 Months					
Groups		G1	G2	G3	G4
	MTBE exposure	Non exposed	MTBE 60 µL/3 min/day/for 3 months	MTBE 60 µL/3 min/day/for 6 months	MTBE 60 µL/3 min/day/for 12 months
Results					
Figures		(Figure 1a)	(Figure 1b–d)	(Figure 1e,f)	(Figure 2a–f)
Tracheal lumen “Lu”		Clear	-	-	Epithelialization Epi
			Necrotic cells “NC”		
Mucosal layer: Lining “respiratory “epithelia “RE”		Normal	Ulceration “MUl”	Desquamation “Ds”	
			Deciliation “Dc”		Flattened “FE”
			Polyp formation PF	Degenerative epithelium “DE”	
			---	Hydropic degeneration “HD”	
			Hyperplasia “Hp”		
			Metaplasia “Mp”		
Lamina propria “LP”		Normal	---	Tracheal adenomas “TA”	
			Inflammatory cells “IF”,		
			Congested blood vessels “CB”,		
			edema “Oe”,		
			Fibroid changes “Fi”		
			Foam cells “FC”		
Hyaline cartilage “HC”		Normal	Perichondrial thickening “PeT”		
			Deformation “Df”	Increase in diameter “Di”	Di with Degeneration DHC
Peritracheal connective tissue “CT”		Normal	---	---	Foam cells “FC”, fatty degeneration “FD”
			---	---	Monocytic infiltration “M”
			Edema “Oe”		
			Fibroid changes “Fi”		
Blood vessels of CT		Normal	Dilated and congested blood vessels “CB”		
Thyroid gland		Normal	----	----	Degenerated thyroid follicles “DT”

**Table 3 biology-09-00002-t003:** Of Histological Alterations of Rat Lung Represented in Figure 3 and Figure 4.

Lung of Rats Exposed to MTBE Vapor Inhalation 60 µL/3 min/day for 3, 6, or 12 Months					
Groups		G1	G2	G3	G4
	MTBE exposure	Non exposed	MTBE 60 µL/3 min/day/for 3 months	MTBE 60 µL/3 min/day/for 6 months	MTBE 60 µL/3 min/day/for 12 months
Results					
Figures		(Figure 1a)	(Figure 3b–d)	(Figure 3e,f)	(Figure 4a–f)
Lumen of Bronchioles “Br”		Normal, Clear	Dilatation in some bronchi “DB”		
Lining epithelia of bronchioles “Br”		Normal	deciliation “Dc” and shortening		---
			---	Metaplasia “Mp”	
			---	Polyp formation PF	Hydroid degeneration “HD”
Peribronchiolar “Pb” and perivascular “Pv” connective tissue “CT”		Normal	---	---	Pulmonary fibrosis Fi
			---	---	Foam cells “FC”
			---	---	Steatosis “St”
			Lymphocytic infiltrations “If”	
			large lymphoid hyperplasia “LH”	
			Edema “Oe”		
			fibroid changes “Fi”	
Lumen of air alveoli		Normal, Clear	---	---	Collapsed alveoli CA
			---	---	Dilated lumen with Epithelialization Epi
			Hemorrhages “H”		H + Diffuse “DH”
Epithelia of air alveoli “AA”		Normal	Emphysematous changes “Em”		Severe Em
			Desquamation “Ds”		
			Shortening with deciliation “Dc”	Degenerated (De) in some air alveoli (Da)	De, Da, metaplasia “Mp”, or thickening “Th”
			---	---	Polyp’s formation “PF”
Interalveolar septa of air alveoli “IA” (Interalveolar CT) and perivascular CT		Normal	Destructed septa “SD”		
			Inflammatory cells infiltrations “If”		If with monocytic infiltration “M”
			Destructed septa “DS”, sever dilated (DV), congested vessels (CB), & capillaries (CC)		Lymphoid hyperplasia “LH”
					Adenomatous changes (lung adenoma) “LA”
					Abscess with central liquefaction “CL” covered by pyogenic membrane “PM”
			Fibroid changes “Fi”	“Fi” with diffused fibrocytes “F”	steatosis (St) with Fi
			---	---	Numerous lipid-laden macrophages “Foam cells” “FC”
			----	----	Mitotic figure “MF”
			Focal hemorrhages “H”		H + Diffuse DH
Blood vessels of CT “BV”		Normal	Dilated vessels “DV”		Occluded lumen
			Congested blood vessels “CB “, Dilatated and Congested blood capillaries “CC”		
Arterial alterations in tunica intima “TI”, tunica media “TM”, and tunica adventitia “Ad”,		Normal TI, TM & Ad	---	---	Desquamation “Ds” in TI
			---	---	Thickening TM & Ad

**Table 4 biology-09-00002-t004:** The molecular weight value for each band on SDS PAGE. Lane M: protein marker, Lane C: control sample. Lane 1: Treated samples after 3 months—Lane 2: Treated samples after six months—Lane 3: Treated samples after 12 months—Lane 4: Treated samples after 12 months.

MW Values	Lane M	Lane C	Lane 1	Lane 2	Lane 3	Lane 4
**Band 1**	70	87	87	87	87	87
**Band 2**	51	82	82	82	82	82
**Band 3**	42	70	77	70	70	70
**Band 4**	32	60	70	60	60	60
**Band 5**	25	51	60	51	51	51
**Band 6**	*****	41	51	41	41	41
**Band 7**	*****	39	41	30	37	37
**Band 8**	*****	25	29	29	29	29
**Band 9**	*****	*****	28	28	28	28
**Band 10**	*****	*****	25	25	25	25
**Band 11**	*****	*****	21	21	21	21
**Band 12**	*****	*****	18	*****	18	18
**Band 13**	*****	*****	*****	*****	*****	*****
**Band 14**	*****	*****	*****	*****	*****	*****
**Band 15**	*****	*****	*****	*****	*****	*****
**Band 16**	*****	*****	*****	*****	*****	*****
**Band 17**	*****	*****	*****	*****	*****	*****

Table 5. The relative mobility for each lane in SDS PAGE. Lane M: protein marker, lane C: control sample. lanes, lane 1: Treated samples after 3 months—Lane 2: Treated samples after six months—Lane 3: Treated samples after 12 months—Lane 4: Treated samples after 12 months.

**Table 5 biology-09-00002-t005:** The relative mobility for each lane in SDS PAGE.

Total RF	Total MW (M)	Lane C	Lane 1	Lane 2	Lane 3	Lane 4
0.04	87	+	+	+	+	+
0.07	82	+	+	+	+	+
0.1	77	−	+	−	−	−
0.15	69	+	+	+	+	+
0.18	65	−	−	−	−	−
0.22	60	+	+	+	+	+
0.28	53	−	−	−	−	−
0.3	51	+	+	+	+	+
0.38	43	−	−	−	−	−
0.41	41	+	+	+	+	+
0.43	39	+	−	−	−	−
0.46	37	−	−	−	+	+
0.51	33	−	−	−	−	−
0.57	29	−	+	+	+	+
0.61	28	−	+	+	+	+
0.66	25	+	+	+	+	+
0.72	21	−	+	+	+	+
0.81	18	−	+	+	+	+

**Table 6 biology-09-00002-t006:** Top 5 Most Abundant Proteins in Each Band Recovered from Mass Spectrometry.

Gene ID	Symbol	Relative Protein Amount (iBAQ)		
		Upper	Middle	Lower
310218	Car1	23172	51634	
54231	Car2	21187	1146	
296973	Bpgm	6293	6543	
24440	Hbb	2678	6378	
25419	Crp	2035	3056	
29338	Prdx2			19904
25475	Lgals5			9310
24786	Sod1			5556
360678	Arhgdia			1590
117254	Prdx1			1022

## Data Availability

The mass spectrometry data for cross-linked EGFR complex identification have been deposited via the MASSIVE repository (MSV000083997) to the Proteome X change Consortium (http://proteomecentral.proteomexchange.org) with the dataset identifier PXD014303).

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
