# Peer review of "Impact Effect of Methyl Tertiary-Butyl Ether “Twelve Months Vapor Inhalation Study in Rats”"

_biology, 2019, doi:10.3390/biology9010002_

Round 1

Reviewer 1 Report

The title of the paper is very misleading. I think based on the information presented through the manuscript, the best conclusion that could be made is that MTBE exposure leads to different levels of protein secretion. And this conclusion is directly related to number 2. What were the control rats exposed to? This information is important so it should be included in the manuscript, as the control needed to be some ether without carcinogenic effect, but with same increase in the inflammatory response. It is not clear why only 5 animals per group were sacrificed at one time. It is not clear for how long per day were the mice exposed to MTBE. The whole experimental design needs more details. A table showing all the identified proteins should be included, together with the relative quantification. As presumably some of the identified proteins are well characterized, they should be used as authentic standards for comparison on gel slides and LC-MS.

Author Response

Response to Reviewer 1 Comments

Point 1: The title of the paper is very misleading. I think based on the information presented through the manuscript, the best conclusion that could be made is that MTBE exposure leads to different levels of protein secretion. And this conclusion is directly related to number 2.

Response 1: Title has been changed to

Impact Effect of Methyl Tertiary-butyl Ether

"Twelve Months Vapor Inhalation Study in Rats"

=============================

Point 2: What were the control rats exposed to? This information is important so it should be included in the manuscript, as the control needed to be some ether without carcinogenic effect, but with same increase in the inflammatory response.

Response 2: This comment considered as very important advice from the reviewer, but we already finish the experimental work. In future studies we should design a special rat group to be used control negative and/or control positive.

=============================

Point 3: It is not clear why only 5 animals per group were sacrificed at one time. It is not clear for how long per day were the mice exposed to MTBE.

First question: clear why only 5 animals per group

Response 3: We use 5 rats per group for these points:

At the beginning of the experimental work, we cannot expect the exact effect of MTBE inhalation histologically. Because this research was funded, we decided that, we should examine most of the tissues especially tracheal and lung in order to prevent missing any evidence, which effect on our responsibilities about the gained results. so we this number of rats. Finally, We essentially need rat blood for serum electrophoresis. From each rat, We get one or two samples from their heart and some blood samples were neglected due to un-known reasons such as hemolysis. Thus, the accurate samples reduced mostly into three samples represent three rats from 5 rats.

Point 4: Second comment: It is not clear for how long per day were the mice exposed to MTBE. The whole experimental design needs more details.

Response 4: Please, note that our experiment on rats not on mice.

We add the following details to the manuscript in the section of Materials and Methods, as shown below:

Experimental design (four groups, n=15)

Rat Groups:

MTBE Vapor dose

Inhalation time

Experimental period

G1: control

Non exposed

Non exposed

0-12 moths

G2: expose to

60 ul/day

3 minutes

0-3 months

G3: expose to

60 ul/day

3 minutes

0-6 months

G4: expose to

60 ul/day

3 minutes

0-12 months

=============================

Point 5: A table showing all the identified proteins should be included, together with the relative quantification.

Response 5: It has been added please refer to table 6

Table 5. Top 5 most abundant proteins per each band recovered from mass spectrometry

Gene ID

Symbol

Relative protein amount (iBAQ)

Upper

Middle

Lower

310218

Car1

23172

51634

54231

Car2

21187

1146

296973

Bpgm

6293

6543

24440

Hbb

2678

6378

25419

Crp

2035

3056

29338

Prdx2

19904

25475

Lgals5

9310

24786

Sod1

5556

360678

Arhgdia

1590

117254

Prdx1

1022

=============================

Point 6:  As presumably some of the identified proteins are well characterized, they should be used as authentic standards for comparison on gel slides and LC-MS.

Response 6: Yes, I certainly agree with the respected reviewer that is why we added Table 5

=============================

Reviewer 2 Report

Nice work. However, it needs to address the attached suggestions. Thank you

Author Response

Response to Reviewer 2 Comments

Point 1: Summary section doesn . For example, sentences

in Line #20 and #24 are incomplete, that need to be paraphrased for better flow the information.

Response 1: It has been corrected

=============================

 Point 2: Also authors claims about the cost effectivity about their protocols compare to current blood-based test however, can author make comments on time and other parameters as these are important in clinic as well.

Response 2: This sentence has been deleted

=============================

 Point 3: How about the possibility of non-invasive biomarker following the methods other than blood (invasive)?

Response 3: Although I respect his question, but I have miss understanding about this comment.

However, I think that:

Invasive biomarker could be (biopsy) histopathological study to detect particular abnormalities, while a non-invasive biomarker gained easily without harm or "pain"

Are the reviewer means that,

If so: the main objective of the present study is to determine severe anomalies such as pro-cancer or stage 1 of cancer than biopsy, in parallel with blood sera test for detect abnormal blood proteins (by electrophoresis). If not: (May be) The reviewer ask about symptoms (non-invasive) on rats after inhalation doses: Some non-invasive symptoms observed, such as noticeable hair loss, short breathing, movability reduced, fighting between them reduced.

=============================

Point 4: Abstract section needed to concise with keeping major finding.

Response 4: This has been corrected please refer to abstract section

=============================

Point 5: A sentence is important at the end of abstract to foresee the future direction of

the finding instead of blunt ending

Response 5: This has been corrected please refer to abstract section

=============================

Point 6: Line #100 NAA is a metabolite, so it should be elevated instead of saying

overexpressed as expression terminology goes well to Gene

Response 6: This has been corrected please refer to line 100 in red color

=============================

Point 7: Overall several nice observations but very dense in manner of presentation. It will

be nice to have a table with all the MTBE doses, observe tissue types and tissue

anomaly s. It will be well understandable if author can just through table like that

besides the text that already exist.

Response 7:

It has been added please refer to table 1 and 2, In addition, the explanation of histology figures re-writed in gradual order corresponding to tissue layers of the trachea and components of lung parenchyma. Each histological anomaly in each layer supported with figure(s) number(s) to prevent any confliction.

=============================

Point 8: Line#504-line#531, this section may need to shorten by removing too much

details of CAs, as the paper need to mostly focus on MTBE and its impact on

health. I think author did nice job in Introduction section and can expand that in

here with new finding.

Response 8: It has been corrected and squeezed please refer to lines 504-531

=============================

Point 9: According to NCI, PSA levels of blood is very controversial for PCa diagnostics,

how can CA in combination of PSA will provide better prognosis?

Response 9: Yes, I agree with the respected reviewer

=============================

Point 10: Line #544 high upregulation, please remove redundant wording

Response 10: It has been corrected

=============================

Point 11: More information about peroxiredoxin 2

Response 11: I added a short paragraph in introduction

Peroxiredoxin-2 is a protein that in humans is encoded by the PRDX2 gene. This gene encodes a member of the peroxiredoxin family of antioxidant enzymes, which reduce hydrogen peroxide and alkyl hydroperoxides. The encoded protein may play an antioxidant protective role in cells, and may contribute to the antiviral activity of CD8(+) T-cells. This protein may have a proliferative effect and play a role in cancer development or progression[17].                         

Point 12: Rewrite discussion

 Response 12: It has been done please refer to discussion section

=============================

Point 13: Line #92 line#98 having error in font sizes

Response 13: It has been corrected

=============================

Point 14:  Line #170 line#173 having error in font sizes

Response 14: It has been corrected

=============================

Round 2

Reviewer 1 Report

The included revisions answered all the questions.

This manuscript is a resubmission of an earlier submission. The following is a list of the peer review reports and author responses from that submission.